# Dental Paleobiology in a Juvenile Neanderthal (Combe-Grenal, Southwestern France)

**DOI:** 10.3390/biology11091352

**Published:** 2022-09-14

**Authors:** María Dolores Garralda, Steve Weiner, Baruch Arensburg, Bruno Maureille, Bernard Vandermeersch

**Affiliations:** 1Departamento de Biodiversidad, Ecología y Evolución, Facultad de CC. Biológicas, Universidad Complutense de Madrid, 28040 Madrid, Spain; 2Department of Chemical and Structural Biology, Weizmann Institute of Science, 234 Herzl Street, Rehovot 76100, Israel; 3Department of Anatomy, Faculty of Medicine, Tel-Aviv University, Tel Aviv 39040, Israel; 4University Bordeaux, CNRS, MC, PACEA UMR5199, F-33600 Pessac, France

**Keywords:** Neanderthal, Combe-Grenal, juvenile, mandible, periodontitis, tooth, tartar, SEM analysis

## Abstract

**Simple Summary:**

Numerous prehistoric sites in Europe and the Near East provided bones and dental remains of the populations of the past. One of them is the Combe-Grenal Cave (SW France), where fossils of children and adults represent the Neanderthals who lived there more than 60 ky ago, during a harsh period of the last glaciation. In this paper, we analyze a sample of the tartar of a juvenile individual. The numerous bacteria forming the plaque are compared to those of one adult from Israel, Kebara 2, revealing the differences between the most common bacteria in a young and an older individual, probably because of their immunological systems, and the different living conditions of the human groups they represented.

**Abstract:**

Combe-Grenal site (Southwest France) was excavated by F. Bordes between 1953 and 1965. He found several human remains in Mousterian levels 60, 39, 35 and especially 25, corresponding to MIS 4 (~75–70/60 ky BP) and with Quina Mousterian lithics. One of the fossils found in level 25 is Combe-Grenal IV, consisting of a fragment of the left corpus of a juvenile mandible. This fragment displays initial juvenile periodontitis, and the two preserved teeth (LLP4 and LLM1) show moderate attrition and dental calculus. The SEM tartar analysis demonstrates the presence of *cocci* and filamentous types of *bacteria*, the former being more prevalent. This result is quite different from those obtained for the two adult Neanderthals Kebara 2 and Subalyuk 1, where more filamentous *bacteria* appear, especially in the Subalyuk 1 sample from Central Europe. These findings agree with the available biomedical data on periodontitis and tartar development in extant individuals, despite the different environmental conditions and diets documented by numerous archeological, taphonomical and geological data available on Neanderthals and present-day populations. New metagenomic analyses are extending this information, and despite the inherent difficulties, they will open important perspectives in studying this ancient human pathology.

## 1. Introduction

The Combe-Grenal site is located east of the Domme village (Dordogne), on the right side of the valley of a small Dordogne dried river. The site, facing south-west, corresponds to a probably very deep and large rock-shelter, naturally formed within Cogniacian limestone. Only a small part of the rock-shelter remains, preserving only a narrow and very small cave at its northeastern angle. Such geomorphology describes the site name “grotte de Combe-Grenal” since at least 1817. Combe-Grenal rock-shelter was first excavated by D. Peyrony in 1929, who identified three Mousterians layers, but the most important investigations were those led by F. Bordes, between 1953 and 1965 [1,2,3,4]. Since 2014, new scientific fieldwork is in progress under the direction of J.-Ph. Faivre (PACEA, Bordeaux).

Considering human fossil remains, several pieces were found by Bordes in Mousterian levels 60, 39, 35 and 25. Geological and faunal studies conducted by Guadeli and Laville [5] attribute level 60 to MIS 6, level 39 to MIS 5a, while levels 35 to 25 are related to MIS 4. Most of the Combe-Grenal fossils (those from layers 35, and especially 25, should be placed chronologically at the beginning of MIS 4 (~70 to 60 ky) and assigned to its coldest period (~70 to 65 ky). Paleoenvironmental and chronostratigraphic data document climatic changes toward colder conditions, first humid and later increasingly drier, as well as a progression of the open Arctic milieu fauna, confirming the cold and harsh environment in which people then lived [5].

The anthropological fossils found at Combe-Grenal were the objects of detailed morpho-anatomical descriptions and analyses [6,7,8,9]. The whole sample is presently preserved at the Musée National de Préhistoire at Les Eyzies (Southwestern France).

Most of the remains were found in level 25, where several young adult males and females (MNI ~8) of different ages were identified [6]. Bordes’ unpublished data demonstrate the dispersion of the fossils in several excavation grid squares, very close to one another, located at the center of the back part of the rock-shelter [6]. All the human fossils were fragmented and randomly mixed with abundant faunal remains and lithics. There were no traces of deliberate burials, but cut marks were identified on several fragments [6,7,9]. Morphological and anatomical analyses of the Combe-Grenal fossils have led to their assignment to Neanderthals.

The aim of this contribution is the study of a tartar sample obtained from the mandibular fragment Combe-Grenal IV. We will briefly summarize the interest in dental calculus analyses, followed by the main morphological characteristics of the fossil, the oral pathology and the results of the tartar SEM analysis in comparison with the previously published data from other Neanderthals and new methods of analyses.

## 2. Brief Reminder of Tartar (Dental Calculus) Etiology

Tartar or dental calculus is a form of hardened dental plaque, caused by the precipitation of minerals from saliva and gingival crevicular fluid in the tooth’s plaque. Such a process kills the bacterial cells within the dental plaque, forming a rough and hardened surface ideal for further plaque formation, namely tartar [10].

Two types of dental calculus have been described. Supragingival tartar affects the gums along the gumline, while subgingival tartar forms within the narrow sulcus existing between the teeth and the gingiva [10]. Dental calculus formation is associated with several clinical manifestations, including receding gums and chronically inflamed gingiva.

According to Lang et al. [10], tartar is composed of both inorganic (mineral) and organic (cellular and extracellular matrix) components. The cells within the dental calculus are primarily bacterial, but also include at least one species of *Archaea* (usually called “*cocci*”) and several species of yeast. Trace amounts of dietary and environmental micro debris or plant DNA have also been found.

The processes of dental calculus formation are not well understood. Tartar forms in incremental layers, but the timing and triggering of these events are poorly understood and vary widely among individuals, probably related to age, gender, diet, etc. [10]. Supragingival tartar is more abundant on the buccal surfaces of the upper maxillary molars and the lingual surfaces of the mandibular molars, while subgingival tartar forms below the gumline and is typically dark in color due to the presence of black-pigmented bacteria.

Dental calculus has been described in animals (e.g., [11]) and documented in various human groups and individuals from Prehistory to present times. Sometimes, even if exceptional, tartar can be present as a very thick deposit, such as on the T15 individual from the Medieval cemetery from Clarensac (Gard, Southeastern France; [12,13]). Concerning human fossil teeth, unfortunately, in the past, many of them have been excessively cleaned, destroying and removing tartar deposits. However, we do have a few with preserved tartar deposits. This is the case for several Early Upper Pleistocene Neanderthals, whose analyses offer new data on their biology and expand the knowledge of their hunter–gatherer population behaviors.

## 3. Materials and Methods

### 3.1. Materials

Combe-Grenal IV is a fragment of the left side mandibular corpus corresponding to the upper part of the left mandibular body (Figure 1), ranging from the distal margin of the left lower second premolar (from here LLP3) alveolus to the mesial septum of the left lower second molar (LLM2; [6]). It preserves two lower teeth, the lower left second premolar (LLP4) and the first molar (LLM1), both with tartar deposits (Figure 1, red arrows) around the crowns and interproximal facets with the LLP3 and the LLM2 (both absent), which indicate that all four teeth had emerged and were functional.

On this mandibular fragment, Combe-Grenal IV, we can also observe the alveolus of the LLM2 (Figure 1C,D). It has highly visible osseous trabeculae, and the alveolar ridge is altered by gingivitis (Figure 1 blue arrows), indicating that the LLM2 had fully erupted and was functional long before the individual’s death, causing the distal interproximal facet on the LLM1. The LLP4 has the mesiodistal axis slightly oblique in comparison to that of the molar. The crown has traces of occlusal attrition (degree 1; [14]) and two interproximal facets, while the mesial one has a deep vertical sulcus [6].

Few age markers can be considered regarding this incomplete fossil, such as the close apex of the two preserved teeth (Figure 1E), their weak attrition (degree 1 from Murphy; [14]) or the small interproximal facet caused on the LLM1 distal side (Figure 1D) by the (absent) LLM2. It is possible to estimate the age at death of this individual by using the charts on dental eruption published for modern children, although they are based on samples with very different biological and environmental conditions. Thus, according to the Ubelaker [15] schemes and the AlQahtani et al. [16] atlas, the age was 15 ± 3 years and 15.5 years, respectively. Consequently, Combe-Grenal IV can be assigned to a juvenile individual.

The alveolar arch (Figure 1 green arrows) displays a slight degree of resorption and is separated from the tooth cement–enamel junction by 2.5/3.02 mm. The loss of osseous mass and the alveolar destruction can also be seen in the radiograph (Figure 1E), showing a pathology that seems to be the result of incipient periodontal disease. This slight exposure of roots, especially if we consider the lack of conjunctive tissue, varies between 1.0 and 1.5 mm in height [17]. Both preserved teeth on Combe-Grenal IV show supragingival tartar deposits (Figure 1 red arrows) forming a wide band around the crowns, separated by 5/7 mm from the alveolar margin. As in degree 2 on the Brothwell [18] scale, and, according to the classification of periodontal diseases [19], they correspond to stage I (early–mild) and nearly stage II (moderate).

Two pulp stones (pulpoliths) of different sizes (that rattle when shaking the fragment) also appear in the pulp chamber of those teeth (Figure 1E, yellow arrows). Such pathology has been documented in other Neanderthals, as in the case with Combe-Grenal X and 29 [6] and Kebara 2 [20,21].

### 3.2. Methods

A small sample of supragingival dental calculus was detached from the lingual surface of the Combe-Grenal IV LLM1 and processed for a scanning electron microscope study at the Weizmann Institute of Science at Rehovot (Israel).

We measured the *bacteria* directly from the micrographs, keeping in mind that these are only estimates. This is because we cannot know whether we are viewing the real length or diameter, given that they were partially embedded in the calcified matrix. The average dimensions were photographed in three pictures, with the different magnifications indicated in Table 1 and Table 2. We measured the *bacteria* appearing more complete (avoiding the empty cavities). Each crown measurement was repeated on three different days by two of the authors (S.W. and B.A.), and the inter- and intra-observer error between measurements was <4%.

On the obtained images, the bacteria identification and measurements were performed by Image Tools 3 “UTHSCA” analysis, and the results are expressed as means ± SE (standard error). The analyses included a breakdown and one-way ANOVA tests. The *p*-values indicated the post hoc significance levels for the respective pairs of means, and a *p*-value of <0.05 was considered significant. The calculations were performed using the SPSS statistical package (1990) and the STATISTICA package (StatSoft Inc., Tulsa, OH, USA, 1995), and their results are given in Table 3.

## 4. Results

The two preserved teeth on Combe-Grenal IV, left LLP4 and LLM_1_, show supragingival dental calculus deposits (Figure 1 red arrows) forming a wide band around the crowns, separated 5/7 mm from the alveolar border, as in degree 2 on the Brothwell [18] scale and I/II of the recent classification [19].

On the surface of the Combe-Grenal IV tartar sample, fine crystal dental calculus deposits appear in the macro photographs at different magnifications. They reveal alternate layers running from the first calcified plaque directly covering the enamel surfaces to the outermost and final calcified layer, indicating various stages in the dental calculus formation of this individual. Magnifications of 3500 µm (Figure 2A), 10,000 µm (Figure 2B) and 20,000 µm (Figure 3A) clearly show both empty bacterial cavities and complete bacteria embedded in the calcified matrix. The bacteria present are *cocci* and filamentous types, although it is not possible to recognize the specific fossilized micro-organisms among the ~325 species that could be present in the oral cavity [22].

We used the SEM images to compare the distribution of these microbiotas in two different Neanderthal fossils: the juvenile Combe-Grenal IV and the adult male Kebara 2 (Israel), for which we have the analysis of a sample also from the calculus of his LLM1 [20].

As can be observed in Figure 3B, the Levantine fossil contains numerous *cocci* bacterial types [20], and rods are more frequent than in the young Combe-Grenal individual.

There are also some differences in the size of the identified bacteria between both individuals. In Table 1 and Table 2, the parameters corresponding to the length and diameter (in µm) of the *bacteria* in the compared fossils cited above are given. They were measured on the SEM microphotographs corresponding to the indicated numbers (Combe-Grenal IV: 304, 306, and 308; Kebara 2: 601, 603, and 606).

In Figure 4, the “Box and Whisker Plot” shows the differences obtained for the total dimensions of Combe-Grenal IV and Kebara 2 *bacteria*, with the former (Figure 4, left) corresponding to the diameter and the latter (Figure 4, right) to the length. On both graphs appear the mean values and the variation range of ± 1 and 1.96 standard errors. The differences between the total values obtained for both fossil individuals (Table 3) are statistically significant, particularly those comparing the diameters (*p* < 0.001), indicating larger dimensions of the bacterial flora on the adult male Kebara 2 than on the juvenile Combe-Grenal IV.

These results agree with the available information on the older adult Subalyuk 1 (Hungary), which—following Pap et al. [23]—shows the presence of more filamentous type *bacteria* than on Kebara 2 [20]. As their dimensions were not given, no statistical comparison with the other two Neanderthals can be made.

## 5. Discussion

Tartar, being a mineralized form of dental plaque adhering to the surface of the tooth, can be preserved, and the study of this durable material using SEM provides information on the microbial flora responsible for the periodontal disease of ancient hominid fossils. It is well known that the presence of periodontitis, in general, is the result of a dense accumulation of micro-organisms on the tooth surface and the host response (innate and acquired immunity) of each individual [24].

The most ancient case among hominids is that described by Ripamonti [25] on a juvenile *Australopithecus*. Concerning the Neanderthals, the publications on the *bacteria* found in the dental calculus of Kebara 2 [20,21], Subalyuk 1 [23] and the present paper demonstrate the rich oral flora present in ancient human populations.

The results of the tartar macroanalyses of these three Neanderthals, Combe-Grenal IV, Kebara 2 and Subalyuk 1, also indicate differences in the oral flora causing supragingival dental calculus and periodontitis among them. When interpreting these findings, it is important to consider not only the chronology and environment in which they lived, but also the individual’s age at death (more *cocci* on the juvenile) and the available data on diets. The latter is difficult to obtain when dealing with sites excavated long ago. Another influencing factor could be immunological differences between individuals, because each one of the three Neanderthals studied could have had different reactions to similar stress, illness or diet. This hypothesis is, however, impossible to verify.

As previously indicated, the three considered Neanderthals have similar antiquity, related to MIS 4 (both European fossils) or the beginning of MIS 3 (Kebara 2), meaning that they lived in a cold and dry environment, undoubtedly colder in Europe than in the Levant. In the case of Combe-Grenal level 25, where the studied fossil was found, sedimentological analyses indicate very cold and dry weather conditions with an open Arctic milieu fauna [5]. Layer 11 from Subalyuk Cave displayed similar conditions, according to Bartucz et al. [26]. In Kebara Cave, the analyses demonstrated that Unit XII (where the burial of the individual Kebara 2 appeared) was formed in a dry and cold environment, but not as extreme as that known for the European sites [27].

None of the three individuals under consideration have been analyzed to obtain the “carbon and nitrogen isotopic signature” used to evaluate aspects of the diet, but some data are available for other Neanderthals found, for example, in Scladina [28], Marillac [29], Vindija [30], Saint-Césaire [31,32], Jonzac [33] or Troisième caverne from Goyet [34]. The results obtained from these fossils indicate that animal tissue must have been a very important source of food, regardless of their different chronologies and environments (e.g., the reindeer so frequent in Marillac, and absent from the Kebara environment). Recently, Ca isotopes were used to assess aspects of the diet of the MIS 5 Neanderthal Regourdou 1 [35]. This study demonstrates the carnivore-like diet of the fossil and, also, the ingestion of a small percentage of bone, probably during the consumption of bone fat and red marrow.

The evaluation of the possible ingestion of vegetables in Paleolithic times is often difficult because of the absence of such remains in many sites, or the imitations imposed by the archeological data sets. Nevertheless, several well-documented excavations, such as those for the Kebara Cave, document not only ephemeral seasonal hunting during late spring–summer, to intensive winter to early spring hunting, but also archeobotanical remains indicating that plant foods appear to have been gathered during the autumn (October–December) and early spring (March–April) and remained more or less constant throughout the analyzed sequence [36,37]. Hardy and colleagues’ [38] research on dental calculus from five El Sidrón Neanderthals has demonstrated the ingestion by these individuals of different kinds of probably cooked plants. Moreover, based on nitrogen isotope analyses for the Spy Neanderthal bone collagen, it was hypothesized that plant consumption accounted for up to 20% of the sources of their diet [39].

The studies on Combe-Grenal IV, Kebara 2 and Subalyuk 1 also allow us to identify the presence of mild periodontitis in Combe-Grenal IV [6], and a more advanced process with thicker dental calculus deposits in the adults Kebara 2 [40] and Subalyuk 1 [23]. The presence of such pathologies on a juvenile specimen is not an exception in Middle Paleolithic times; the Combe-Grenal I (~7 years old) child’s mandible shows a slight periodontosis and a thin tartar line on the two preserved deciduous molars [6]. Genetic factors and living conditions are highly correlated to the development of periodontal disease. Note, also, that the marked interproximal facets described on the juvenile Combe-Grenal IV LLP4 and LLM1 [6] reflect strong masticatory forces, which developed daily, perhaps not only due to diet but also due to paramasticatory activities.

## 6. Conclusions

The study of the mandibular fragment Combe-Grenal IV (MIS 4, ~ 70 ky) allows the assignment of this fossil to a relatively robust juvenile Neanderthal individual (~15 years old). Two teeth are preserved (LLP4 and LLM_1_), and show moderate attrition, pulpoliths and initial root hypercementosis. Mild periodontitis affected the alveolar region, and both teeth display almost continuous 1 mm high strip dental calculus on their buccal crown lengths.

The SEM analysis of the LLM1 dental calculus demonstrated the prevalence of *cocci*-type *bacteria*, which is usual in juvenile individuals. This group of microbiota is less prevalent in the tartar from the adult male Kebara 2, where “rods” are more frequent, although less so than in the Subalyuk 1 older adult. In agreement with these observations, the measurements made on the SEM pictures from Combe-Grenal IV and Kebara 2 show larger dimensions of the *bacteria* in the latter individual, with statistically significant differences being found between the means of the lengths and, particularly, between the diameters.

It is interesting to remark that tartar was preserved in the studied fossils, while other individuals, such as the late Neanderthal maxillary CF-1 from Cova Foradà [41], was assigned to an old individual who suffered advanced periodontal disease, but no tartar has been described. In other fossils, such as Krapina-J, the modern manipulations that removed the dental plaque are visible.

Over the last two decades, scientists have relied increasingly on analyses of stable carbon, nitrogen and oxygen isotopes, as well as strontium and other trace elements, in bone, tooth enamel and dentine in order to determine the role of plants, animal tissues and fish in past human diets. Various studies indicate that one can differentiate between the consumption of C3 and C4 plants and trace the exploitation of terrestrial and marine mammals and fish, e.g., [42,43,44]. Even with the use of these techniques, we still cannot determine the ratios of animal tissues versus plant foods when analyzing Neanderthal remains [45,46,47]. However, testing the correlation between ethnographic information and stable isotopes on samples of more recent chronology has demonstrated good agreement [48].

At present, new possibilities are opening with metagenomic analyses which can enable the detailed study of the microbial genomes preserved in the dental calculus. The first results demonstrated, as expected, the regional differences in ecology among the several Neanderthal specimens considered, and consequently differences in diet and paleogenetics [49], and some of them are questionable [50,51]. Similar results have been published by studying the faunal remains found on numerous sites (e.g., [50] in Europe and the Near East, often allowing for the determination of the seasonal periods of hunting, as in the cited case for Kebara [34,35] or Marillac [52]). The macromorphological tartar studies that we present in this contribution for Combe-Grenal IV, or those on the Kebara 2 or Subalyuk 1’s tartar, reflect the development of the microbiota related to the different diets and ages at death of these Neanderthals, evolving through life to include more rods, as is known for extant populations [10].

## Figures and Tables

**Figure 1 biology-11-01352-f001:**
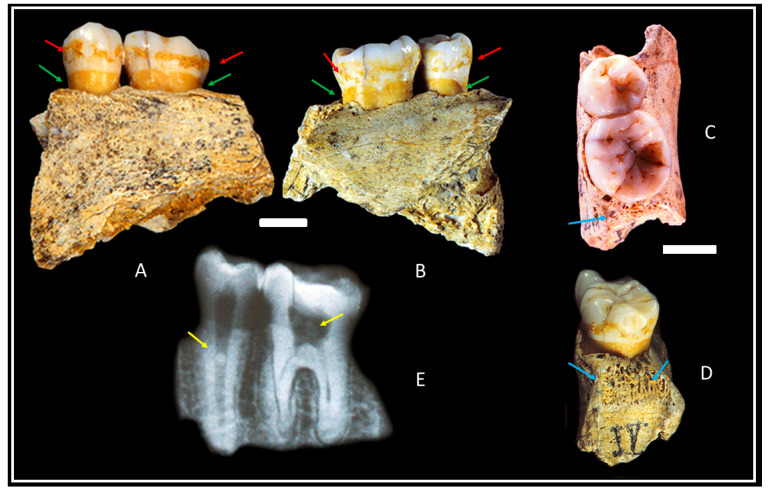
Combe-Grenal IV: external (**A**), internal (**B**), occlusal (**C**), distal side of the LLM1and the alveolus of the LLM2 (**D**), X-ray (**E**) external side. Scales = 10 mm.

**Figure 2 biology-11-01352-f002:**
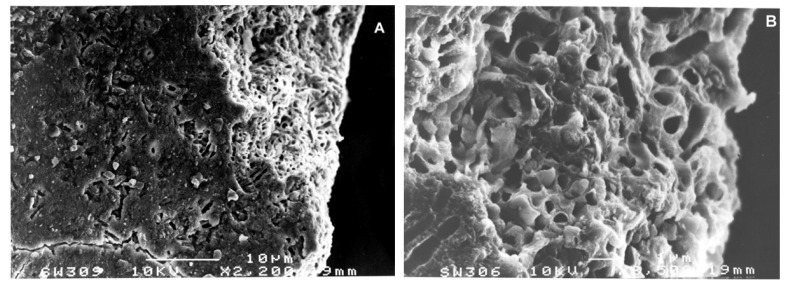
Combe-Grenal IV. (**A**): (picture 306), tartar *bacteria* at 3500 SEM magnification. (**B**): (picture 304): tartar *bacteria* at 10,000 SEM magnification, white scale = 1 μm.

**Figure 3 biology-11-01352-f003:**
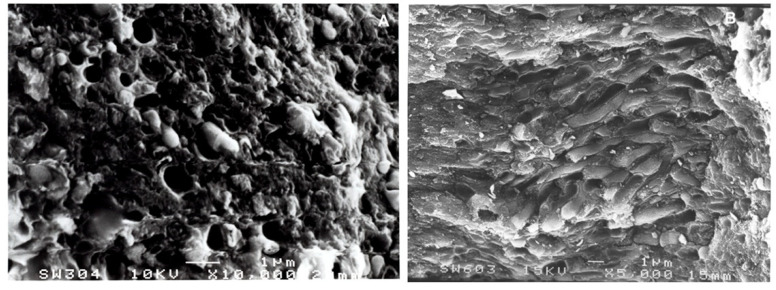
Combe Grenal IV. (**A**): (picture 308), tartar *bacteria* at 20,000 SEM magnification. (**B**): Kebara 2 (picture 603), tartar *bacteria* at 5000 SEM magnification.

**Figure 4 biology-11-01352-f004:**
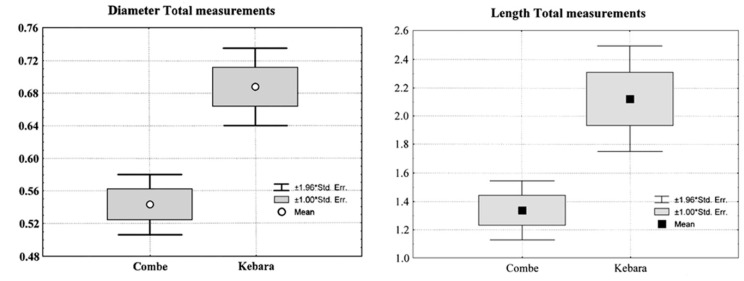
Comparison (Box and Whisker Plot) of the total diameter (**left**) and length (**right**) measurements (in μm) of the Combe-Grenal IV and Kebara 2 bacteria. The mean and the ranges of variation of ±1 and 1.96 error standard deviations are indicated.

**Table 1 biology-11-01352-t001:** Measurements in μm (diameter and length) of the Combe-Grenal IV tartar *bacteria*, taken from the SEM pictures indicated.

	Measurements	N	Mean	Std. Dev.	Std. Error	Minimum	Maximum	Variance
C. Grenal 304 (10.000 M)	Diameter	16	0.493	0.109	0.027	0.360	0.700	0.012
	Length	7	0.791	0.262	0.099	0.590	1.340	0.069
C.-Grenal 306 (3.500 M)	Diameter	31	0.601	0.138	0.025	0.310	0.850	0.019
	Length	10	1.608	0.472	0.149	1.050	2.550	0.222
C.-Grenal 308 (20.000 M)	Diameter	15	0.477	0.165	0.043	0.330	0.840	0.027
	Length	7	1.499	0.376	0.142	1.160	2.200	0.141
Total C.-Grenal IV	Diameter	62	0.543	0.149	0.019	0.330	0.850	0.022
	Length	24	1.338	0.522	0.107	0.590	2.550	0.272

**Table 2 biology-11-01352-t002:** The measurements in μm (diameter and length) of the Kebara 2 tartar bacteria, taken from the indicated SEM pictures.

Pictures	Measurements	N	Mean	Std. Dev.	Std. Error	Minimum	Maximum	Variance
Kebara 601 (10.000 M)	Diameter	20	0.718	0.161	0.036	0.350	0.940	0.026
	Length	18	1.241	0.422	0.100	0.830	2.360	0.178
Kebara 603 (5.000 M)	Diameter	17	0.825	0.154	0.037	0.570	1.140	0.024
	Length	16	3.116	0.719	0.180	1.940	4.310	0.517
Kebara 606 (5.000 M)	Diameter	22	0.555	0.142	0.030	0.400	0.810	0.020
	Length	0						
Total Kebara	Diameter	59	0.687	0.187	0.024	0.400	1.140	0.035
	Length	34	2.123	1.109	0.190	0.830	4.310	1.229

**Table 3 biology-11-01352-t003:** Statistical analysis of the total lengths and diameters of the tartar *bacteria* from Combe-Grenal IV and Kebara 2.

Individuals	Traits	N	Mean	Std. Error	P
Total Kebara 2	Diameter	59	0.687	0.187	0.0001
Total Combe-Grenal IV		62	0.543	0.149	
Total Kebara 2	Length	34	2.123	1.109	0.02
Total Combe-Grenal IV		24	1.338	0.522	

## Data Availability

Not applicable.

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
