# Peer review of "Dental Paleobiology in a Juvenile Neanderthal (Combe-Grenal, Southwestern France)"

_biology, 2022, doi:10.3390/biology11091352_

Round 1

Reviewer 1 Report

The authors present results of bacteriae in tartar formation using SEM analysis of a Neandertal juvenile approximately 15 years of age. 

My first comments relate to the use of LLP4 and 3? Is this a typo should it not be LLP2?  I would also like the authors to consider that what they have identified as gingivitis is also consistent with alveolar resorption resulting from tooth loss (LM2). I would encourage the authors to include this caveat so as to avoid over-interpretation.

More detail is needed in the methods section regarding how the bacteriae were measured. For example, did they measure a section of the image or the entire image? How were the bacteriae differentiated from other potential sources? It would be helpful to the reader to have a figure that would show how the measurements were taken.  Like using osteons for age estimation this method appears may have a large degree of inter and intraobserver error. Was an inter- intraobserver error test conducted? What is being compared with the ANOVA (between teeth)? At what magnification were the measurements taken?

Author Response

In Paleoanthropology we study the evolution of dentition in Primates. Regarding the dentsl formula, the oldest forms had 4 premolars in each maxilla (P1, P2, P3, P4), es e. g. Adapis. Later the ancestral forms lost the P1, and many old and extant Primates (e. g., all the  American Platyrrhini) had 3 premolars in each maxilla (P2, P3, P4). It is only ∼30 My ago that we identified the first Primate (Aegyptopithecus) having only the two last premolars (p3 and P4; that is, the elimination of the two firts premolars), as most recent Hominoidea (among which is the genus Homo). In our papers is very common to speak on P3 and P4, and it is accepted in the best journals of our field (e. g., Wolpoff, 1997), Am. J. of Physical Anthropology; Carbonell et al., 2008, Science; Bermúdez de Castro et al., 2010, PNAS; Martinón-Torres et al., 2012, J. of Human Evolution; Détroit et al, 2019, Nature; Garralda et al., 2019, J. of Human Evolution)

2.- The LLM2 was erupted and in position, causing the distal interproximal facet on the LLM1. In the sediments, and probably by taphonomic processes, the mandible was broken, and we found exclusively this fragment. We don't have any data allowing us to say that the LLM2 shed intra-vitam. The weak occlusal wear on the two preserved teeth corresponds to what is identified in other young Neanderthals.

3.- We corrected and detailed the text on the methods. The magnifications where measurements were taken are indicated (Tables 1 and 2). We detailed the inter- and intra-observed error test.

4.- the magnifications were included in Tables 1 and 2

5.- The numerous SEM pictures and their careful observation permit the identification of the bacteriae or the vide holes in the matrix.

6.- We have done two Anova tests to compare the diametr and length variabilities of the C-Grenal IV and Kebara 2 bacteriae (cf. Table 3) 

Reviewer 2 Report

After reviewing the manuscript “DENTAL PALEOBIOLOGY IN A JUVENILE NEANDERTHAL (COMBE-GRENAL, SOUTHWESTERN FRANCE)” according to the criteria for publication of the journal Biology, we have made ​​the following assessment:

Review Criteria: After a brief bibliographical search, we have found that the lack of studies on dental health of juvenile Neanderthals makes this paper of a great interest. We find especially interesting the findings of the present study in relation to the juvenile periodontitis and the new metagenomic analyses.

The clarity in drafting, and its approach is adequate. The review is exhaustive, and bibliography is representative of the ideas of the paper. With regard to the abstract or keywords, are also adequate and representative of the general idea of the study.

Due to the quality and clarity, our proposal is to publish without modifications.

Author Response

Thak you very much for your interest

Reviewer 3 Report

The paper is interesting and addresses the problem of periodontal disease in Neanderthal. Although the work is of a palaeopathological nature, it would be advisable to use the nomenclature used in modern medical science. As of 2017, the New Nomenclature for Periodontal Diseases, which was agreed at the World Workshop on the Classification of Periodontal and Peri-implant Diseases and Disorders held in Chicago in November 2017, is in force. Therefore, terms used in the publication such as juvenile periodontitis should be changed to the currently stasis terms.

[198] In general, always periodontal disease in humans and animals is the result of the activity and presence of microorganisms on the one hand and the host response on the other. This should be mentioned in the discussion

[253] This is an overgeneralisation, because in humans and animals, genetic factors and risk factors, which include living conditions, are of major importance for the development of periodontal disease. Diet is of secondary importance. This should be discussed in more detail

Author Response

1.- We used the new nomenclature for periodental diseases as far as possible, because we are not dentists, nor do work in clinical practice on extant people

Juvenile periodontitis was eliminated.

2.- We included a few text on the presence of microorganisms in the mouth and the immnunogentics of everyone.

3.- We modified the paragraph.

Round 2

Reviewer 3 Report

After the introduced changes, the paper can be published